# Online randomised trials with children: A scoping review

Simone Lepage[1,2]☯*, Aislinn Conway[3], Noah Goodson[4], Paul Wicks[5], Laura Flight[6], Declan Devane[1,2,3,7]☯

1 Health Research Board–Trials Methodology Research Network (HRB-TMRN), University of Galway, Galway, Ireland, 2 School of Nursing and Midwifery, University of Galway, Galway, Ireland, 3 Evidence Synthesis Ireland, University of Galway, Galway, Ireland, 4 Data & Analytics, Thread Research, Tustin, California, United States of America, 5 Wicks Digital Health, Lichfield, Staffordshire, United Kingdom, 6 National Institute for Health and Care Excellence, Piccadilly Plaza, Manchester, United Kingdom, 7 Cochrane Ireland, University of Galway, Galway, Ireland

☯ These authors contributed equally to this work.
* s.lepage1@universityofgalway.ie

## Abstract

### Background

Paediatric trials must contend with many challenges that adult trials face but often bring additional obstacles. Decentralised trials, where some or all trial methods occur away from a centralised location, are a promising strategy to help meet these challenges. This scoping review aims to (a) identify what methods and tools have been used to create and conduct entirely online-decentralised trials with children and (b) determine the gaps in the knowledge in this field. This review will describe the methods used in these trials to identify their facilitators and the gaps in the knowledge.

### Methods

The methods were informed by guidance from the Joanna Briggs Institute and the PRISMA extension for scoping reviews. We systematically searched MEDLINE, CENTRAL, CINAHL, and Embase databases, trial registries, pre-print servers, and the internet. We included randomised and quasi-randomised trials conducted entirely online with participants under 18 published in English. A risk of bias assessment was completed for all included studies.

### Results

Twenty-one trials met our inclusion criteria. The average age of participants was 14.6 years. Social media was the most common method of online recruitment. Most trials employed an external host website to store and protect their data. Duration of trials ranged from single-session interventions up to ten weeks. Fourteen trials compensated participants. Eight trials involved children in their trial design process; none reported compensation for this. Most trials had a low risk of bias in "random sequence generation", "selective reporting", and "other". Most trials had a high risk of bias in "blinding participants and personnel", "blinding

**Data Availability Statement:** All relevant data are within the paper and its Supporting Information files. Supporting data are also available in an OSF repository: https://osf.io/4w5kq/

**Funding:** This research was supported by the Health Research Board [TMRN-2021-001] and by the College of Medicine, Nursing and Health Sciences at the University of Galway. Funding was received by DD from the Health Research Board of Ireland: https://www.hrb.ie/ The funder had no role in study design, data collection and analysis, decision to publish, or preparation of the manuscript.

**Competing interests:** PW is an associate editor at the Journal of Medical Internet Research and is on the editorial advisory boards of The BMJ, BMC Medicine, The Patient, and Digital Biomarkers. PW is employed by Wicks Digital Health Ltd, which has received funding from Ada Health, AstraZeneca, Biogen, Bold Health, Camoni, Compass Pathways, Coronna, EIT, Endava, Happify, HealthUnlocked, Inbeeo, Kheiron Medical, Lindus Health, MedRhythms, PatientsLikeMe, Sano Genetics, Self Care Catalysts, The Learning Corp, The Wellcome Trust, THREAD Research, VeraSci, and Woebot. NG is an employee of THREAD, a decentralized platform provider. The remaining authors, AC, DD, LF, and SL have no conflicts of interest to report. The findings and conclusions in the document are those of the authors and not necessarily those of NICE. This does not alter our adherence to PLOS ONE policies on sharing data and materials.

of outcome assessment", and "incomplete outcome data". "Allocation concealment" was unclear in most studies.

## Conclusions

There was a lack of transparent reporting of the recruitment, randomisation, and retention methods used in many of the trials included in this review. Patient and public involvement (PPI) was not common, and the compensation of PPI partners was not reported in any study. Consent methods and protection against fraudulent entries to trials were creative and thoroughly discussed by some trials and not addressed by others. More work and thorough reporting of how these trials are conducted is needed to increase their reproducibility and quality.

## Ethics and dissemination

Ethical approval was not necessary since all data sources used are publicly available.

## Introduction

Randomised controlled trials (RCTs) are designed to minimise bias in evaluating healthcare interventions. When well executed, RCTs can provide credible evidence to help researchers determine the effectiveness of an intervention [1]. While the number of trials is growing [2], concerns are consistently raised about the quality of trials [3]. RCTs are often expensive and complex, and paediatric trials bring additional obstacles [4], including added ethical challenges [5], scarcity of funding [5,6], and recruitment barriers [2]. In 2015, a retrospective study of 559 paediatric RCTs found that 19% (n = 104) were discontinued early [6]. A 2022 review of 13,259 trials in people under 18 found that 11% (n = 903) were discontinued before completion due primarily to recruitment barriers [4]. Not only are paediatric trials more challenging to navigate and complete, but fewer are also conducted [2]. In an analysis of 4,146 RCTs, Groff et al. identified 14.2% (n = 591) of trials enrolled paediatric participants, 67.3% (n = 2794) enrolled adults, and 18.3% (n = 761) enrolled both [2]. The U.S. government and European Parliament have issued policies in the last two decades calling for more high-quality paediatric clinical trials to increase medicinal safety and efficacy for children [5,7,8]. Despite these initiatives, paediatric trials are still underfunded [6], conducted less frequently [2], and published less often [4,6].

One strategy increasingly used to address the challenges of conducting randomised trials is decentralising the trial [9–11]. A *decentralised trial* is an umbrella term that captures trial methods that occur away from a central point of trial conduct, i.e., collecting data via multiple mobile clinics instead of one large hospital, leveraging digital technologies (e.g., wearable devices), or moving a trial entirely online [12]. Decentralised trials offer wider-reaching recruitment opportunities [11], diversify a population sample [10], decrease the burden of involvement on participants [12], and research waste and trial costs [11,12]. These trials have existed for many years [9–12]; however, the COVID-19 pandemic has emphasised their potential. Many clinical trials were halted in early 2020, but Price et al. [13] found that by May 2020, participant recruitment in decentralised trials not only recovered from this forced pause but exceeded pre-pandemic recruitment. In a survey of 1,002 rheumatology patients in 2020,

Mirza et al. [14] reported that over 75% of patients responded positively to decentralised interventions such as remote trial visits.

Despite their advantages, researchers and reviewers have expressed apprehension about the ability of decentralised, particularly entirely online, trial methods to produce reliable results [15]. Online trials require careful and critical planning to sample appropriate populations, avoid fraudulent data, and ethically incentivise participants [15]. While decentralised strategies promise to improve trial processes, not all trials lend themselves to complete decentralisation. Partially decentralised or 'hybrid' trials have grown in numbers in the last two decades [9]. Hybrid trials include an aspect of decentralisation combined with traditional methods [10,13].

In this paper, we are interested in fully online RCTs with children to inform the design and conduct of our own online trial with children. Given the absence of guidance for decentralised paediatric trials, this scoping review aims to (a) identify what methods and tools were used to conduct online trials with children and (b) determine the gaps in the knowledge. We describe recruitment methods, consent, data collection, compensation, loss to follow-up, and public and patient involvement (PPI) in online trials within paediatric populations. We also assess the risk of bias (RoB) for each trial and identify facilitators in the conduct of these trials.

## Methods

The Preferred Reporting Items for Systematic Reviews and Meta-analyses extension for Scoping Reviews (PRISMA-ScR) [16], the Joanna Briggs Institute's Manual for Evidence Synthesis [17] and Peters et al. [18] guided the review process. The completed PRISMA-ScR checklist is available in S1 Checklist.

### Protocol, registration, and ethics

The published protocol for this scoping review details our methodology [19]. The study is registered with Open Science Framework (OSF) at (https://doi.org/10.17605/OSF.IO/WHJXY) [20]. Ethical approval was not required as all included data sources are publicly available.

### Information sources, search strategies, and eligibility criteria

**Information sources.** We searched the following sources:

1. Cochrane Central Register of Controlled Trials (CENTRAL),

2. Cumulated Index to Nursing and Allied Health Literature (CINAHL) (EBSCO),

3. Embase (Elsevier),

4. MEDLINE (Ovid).

5. World Health Organization's (WHO) International Clinical Trials Registry Platform (ICTRP)

6. EU Clinical Trials Register

7. NIH Clinical Trials Register

8. medRxiv, JMIR Preprints, HRB Open Research, and Advance from SAGE for pre-prints

9. Google and Google Scholar.

**Search strategies.** A detailed description of how the search strategy was devised is outlined in the protocol [19]. The strategy was constructed by SL and modified and finalised by

an information and knowledge translation specialist (AC). The search strategies for all databases used are available in S1 Appendix. To search trial registries and preprint servers, we used the strategy developed in MEDLINE to guide us iteratively. We used different search terms and string combinations because preprint servers and trial registries do not allow for the same search complexity level as database libraries. The search strategies used for all grey literature searches are in S2 Appendix. We manually searched the reference lists of all studies that met the inclusion criteria. Searches were limited to English-language publications. We did not limit our search by date as detailed in S2 and S3 Appendices.

**Eligibility criteria.** Inclusion criteria were as follows:

1. randomised or quasi-randomised trials

2. participants under 18 years

3. children had to be active participants who contributed data

4. digital technologies connected to the internet had to be the primary method used in each trial phase.

Posters and conference abstracts were excluded due to the likely paucity of methods reporting.

All inclusion and exclusion criteria were pre-determined and detailed in the protocol [19]. One additional criterion was added after the protocol was published to clarify the required age of study participants. Trials where participants were an age-range cohort (e.g., 16- to 25- year-olds) and ≥ 60% were under 18 years (e.g., 100 participants aged 16 to 25, and 65 of them were 18 years or under) were included. If a trial did not specify the exact number of participants under or above 18, the mean age of all participants had to be under 18-years-old.

## Screening and selecting evidence sources

Following the search, all identified citations were imported into EndNote20 [21], the libraries were collated, and duplicates were removed. These titles and abstracts were uploaded into Covidence, a web-based software developed by the Cochrane group to streamline evidence synthesis reviews [22]. We originally planned to use SysRev [23] to screen titles, abstracts, and full texts, but this changed from the protocol during the piloting process. Covidence allowed us to expedite screening, as the two authors were more familiar with this software.

An initial pilot test (performed by SL and DD) screening 50 abstracts resulted in inter-rater reliability (IRR) of 72% using Cohen's kappa [24]. As outlined in the protocol, the target IRR was 75%, so the same two authors completed an additional pilot with 100 abstracts. This second pilot resulted in an IRR of 90%. In the interest of resource conservation and the designated screening author's familiarity with the review's objectives, a single author (SL) screened the remaining titles and abstracts. Two authors (SL and DD) completed a full-text pilot of 50 trials independently with an IRR of 77% calculated using Cohen's kappa [24]. The goal IRR was 75%, and per the protocol, one author (SL) screened the remaining full texts.

## Data charting

Two authors (SL and DD) piloted the original data charting form published in the protocol and modified it to capture all pertinent data appropriately. Because the data charting process for this review was mainly narrative, an IRR was not calculated. To ensure a satisfactory level of agreement between reviewers, they independently charted five reports with the new data charting form, discussed any discrepancies, and came to a consensus. A single author (SL) charted all additional full texts. Data were charted using the primary articles and, where

applicable, the protocols of the trials. Study authors were not contacted for further information. The data charting form can be accessed in S1 Form.

We used the Cochrane Group's RoB1 tool [25] as all included trials were either randomised or quasi-randomised and lent themselves to this assessment. Two authors (SL and DD) completed the RoB assessments independently and in duplicate. We consulted a third author (LF) for one trial with a persistent discrepancy in one domain. As published in the protocol [19], we planned to use RobotReviewer [26], an automated risk of bias assessment tool. We deviated from the protocol here, as using the automation tool and two independent reviewers was redundant.

## Data synthesis

We described the general characteristics, recruitment methods, consent, data collection, compensation, loss to follow-up, public and patient involvement (PPI), RoB, and any findings pertinent to this review in online trials within paediatric populations.

## Results

### Search results

We screened 6,957 titles and abstracts and excluded 6,550 records. We screened 407 full texts. Twenty full texts met the inclusion criteria. These records' references were manually searched, and one additional full text was added, resulting in 21 studies identified that met the inclusion criteria. Fig 1 captures the information sources, search results, justifications for excluding full texts, and identification strategy for relevant studies in the PRISMA flow diagram [27].

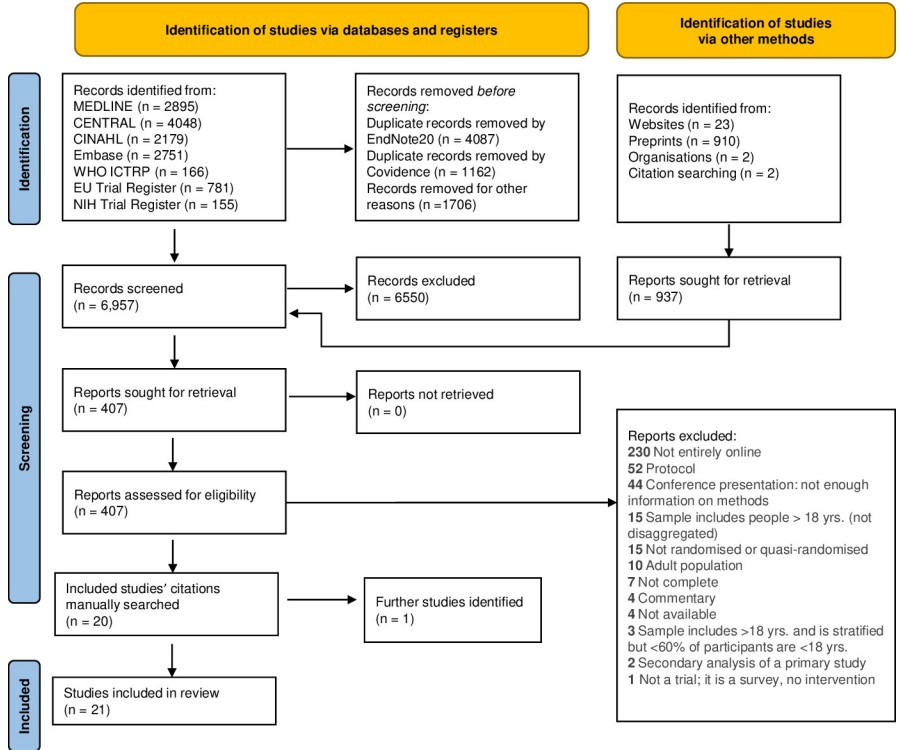

**Fig 1. PRISMA 2020 flow diagram.**

## Characteristics of included studies

Table 1 displays the basic characteristics of all included trials [28–48]. The years of publication of the studies ranged from 2014 to 2022, where the majority (n = 14) were published in the last two years. There were four pilot trials. The number of participants ranged from 45 to 2,452. The pilot trials had, on average, fewer participants (x̄ = 86) than the full trials (x̄ = 560.7). The mean age of children who participated was 14.6 years. Of the 21 included trials, seven were single-session interventions, and one was unclear (Table 1). The remaining 13 trials ranged in duration from three- to ten- weeks.

Because of the breadth of our scoping review, the included studies are heterogeneous in design, aims, and topic of study. We categorised the trials into four groups: mental health (n = 13), substance use issues (n = 2), sexual health (n = 1), and public health and education (n = 5). Four trials collected data contributed by both caregivers and children. Four of the 21 trials reported on gender or sexual minority identification. Two trials intentionally enrolled only female participants, and one intentionally enrolled only males. Sixteen trials reported race or ethnicity (S1 Table). All 16 reported that most participants were White, the largest racial group in their respective countries of conduct [49–55].

## Recruitment methods

Social media was the most used platform for recruiting participants (n = 15), and nine trials relied on it exclusively. Seven trials leveraged multiple methods for their recruitment. Four trials only used external, professional recruitment panels for their online recruitment, and one used email alone. Detailed recruitment methods for all trials are available in S2 Table.

Of the social media platforms, Facebook, or a combination of Facebook and one or more additional social media platforms, was used in 53.3% (n = 8/15) of trials. Seven of the 15 trials (46.7%) using social media used Instagram, exclusively or in combination with others. Four trials did not identify which social media platforms they used.

All seven trials using multiple recruitment methods employed an alternative form of online advertising to social media, i.e., banner ads on different web pages, search engine advertising, live-streaming events, and email listservs. Of these seven trials, six combined other online advertising with social media. Three trials used a dedicated website for recruitment, combined with at least one other method [29,34–35] (S2 Table). Table 2 illustrates the online platforms leveraged by the included trials and the social media platforms used.

Eight trials combined their online recruitment with offline advertising of the trial [29,30, 34,35,39,40,43,46]. Offline recruitment included advertisements in communities, through healthcare professionals, and schools. Offline recruitment details are available in S2 Table.

## Consent

All included trials reported acquiring consent from participants. Parental/caregiver consent was required in twelve of the trials [30,31,35,36,38–41,43–46], waived in five of them [32,34,42,47,48] and not reported in four [28,29,33,37]. Ten trials validated participants' consent via attention-check questions, quality-check questions, ongoing consent, and orientation videos [30–32,35,38,40,42,43,46,48].

Seven trials [28,30,32–34,37,39] sent links to their participants via email or a direct social media message to direct them to an e-consent form. Five trials [41–43,46,47] used an external website (e.g., Qualtrics, REDCap, IRIS) that hosted their e-consent. Participants were automatically redirected through a click-link to an external website for the consent process. Two trials [38,40] acquired consent verbally through videoconferencing, two used a dedicated study website [44–45], and one used a dedicated study app [48]. One trial accepted consent through

**Table 1. Overview of characteristics of included studies.**

| Study (country) | Study type | Number of participants randomised[$] | Age of participants (mean age) in years | Intervention duration | Topic of study | Contribution of data by caregivers | Participants by reported gender % |
|---|---|---|---|---|---|---|---|
| Amsalem and Martin, 2022 [28] (U.S.A.) | RT | 1,183 | 14–18 (16.8) | SSI | Mental health | No | 47.0 female 53.0 male |
| Arnaud et al., 2016 [29] (Belgium, Czech Republic, Germany, Sweden) | RT | 1,449 | 16–18 (16.8) | SSI | Substance use | No | 48.2 female 51.8 male |
| Bragg et al., 2021 [30] (U.S.A.) | RT | 832 | 13–17 (14.7) | SSI | Public health | No | 48.8 female 51.2 male |
| Craig et al., 2016 [31] (U.S.A) | RT | 59 (child/ caregiver dyads) | 7–11 (9.7) | 10 weeks | Mental health | Yes | 41.0 female 59.0 male |
| Dobias et al., 2021 [32] (U.S.A.) | RT | 565 | 13–16 (14.9) | SSI | Mental health | No | A |
| Egan et al., 2021 [33] (U.S.A.) | RT | 240 | 14–18 (15.8) | 4 weeks | Mental health | No | B |
| Ghaderi et al., 2020 [34] (Sweden) | RT | 443 | 15–20 (17.3) | 4 weeks | Mental health | No | 100.0 female 0.0 male |
| Greene et al., 2020 [35] (U.S.A.) | RT | 713 | 12–17 (14.7) | 3 weeks | Substance use | No | 66.0 female 34.0 male |
| Hillhouse et al., 2017 [36] (U.S.A.) | RT | 443 (child/ caregiver dyads) | 12–18 (15.2) | Unclear, likely an SSI | Public health | No | 100.0 female 0.0 male |
| Kelleher, Moreno, & Wilt, 2018 [37] (U.S.A.) | Pilot RT | 45 | 15–23 (17.5) | SSI | Mental health | No | 65.0 female 35.0 male |
| Lester et al., 2019 (Canada, Columbia, U.S.A.) [38] | Pilot RT | 51 | 12–17 (14.4) | 8 weeks | Mental health | No | 41.0 female 59.0 female |
| Manicavasagar et al., 2014 [39] (Australia) | RT | 235 | 12–18 (15.4) | 6 weeks | Mental health | No | 67.5 female 32.5 male |
| Mogil et al., 2021 [40] (U.S.A.) | RT | 200 (families) | 3–6 (4.5) | 4–10 weeks | Public health | Yes | 51.2 female 48.8 male |
| Moreno et al., 2021 [41] (U.S.A.) | RT | 1,520 (child/ caregiver dyads) | 12–17 (14.5) | 8 weeks | Public health | Yes | 51.9 female 48.1 male |
| Nelson et al., 2022 [42] (U.S.A.) | Pilot RT | 154 | 14–17 (16) | 3 weeks | Sexual health | No | 100.0 female |
| O'Connor et al., 2020 [43] (Canada) | Pilot RT | 94 | 13–17 (15.3) | 8 weeks | Mental health | No | 90.0 female 10.0 male |
| O'Dea et al., 2020 [44] (Australia) | RT | 193 | 12–16 (14.8) | SSI | Mental health | No | 86.5 female 13.5 male |
| Parker, Scull, & Morrison, 2022 [45] (U.S.A.) | RT | 132 | 8–14 (10.9) | 3 weeks | Public health | Yes | 49.5 female 50.5 male |
| Radomski et al., 2020 [46] (Canada) | RT | 536 | 13–19 (16.6) | 6 weeks | Mental health | No | 71.3 female 28.7 male |
| Schleider et al., 2022 [47] (U.S.A.) | RT | 2,452 | 13–16 (14.8) | SSI | Mental health | No | 88.1 [c] |
| Schwinn et al., 2015 [48] (U.S.A.) | RT | 236 | 15–16 (16.1) | 4 weeks | Mental health | No | 49.6 [d] |

[$] = number of participants refers to either the number of children OR the number of child-caregiver groups randomised

RT* = Randomised trial.

SSI ** = Single-session intervention.

A = 37.35% of participants in Dobias et al. identified as a gender minority.

B = Egan et al. collected the following data: 16.3% identified as a cisgender girl, 36.7% as a cisgender boy, and 47.1% as a gender minority.

[c] = Schleider et al. reported on 16 gender categories, and 80% of participants identified as a sexual minority.

[d] = Schwinn et al. reported that 18.3% of participants identified as a gender minority.

email, text messages, their dedicated website, and fax and mail [35]. Four trials [29,31,36,40] did not report the methods used to gain consent, stating only that it was acquired, although Craig et al. [31] described the procedures leading up to consent.

## Data collection

Over half of the trials (52.4%, n = 11) used externally hosted websites specialised in data collection. Five trials reported using an externally hosted website and their own internally hosted, dedicated websites. Four trials reported collecting data with their own dedicated study website with no additional external-host platform.

Not all trials employed websites for data collection. Mogil et al. [40] reported capturing their data via videoconferencing without specifying the platform. Four trials [35–37,45] used surveys sent to participants for completion and return, and one study [48] did not specify how they collected their data, only that it was done online. Four trials used validation to protect against duplicate or fraudulent entries to the trial [28,30,37,42]. Validation methods included open-ended questions, CAPTCHA questions, timers (to ensure participants were not simply clicking through the questions), flagging suspicious IP addresses or geographical coordinates, and short completion times.

## Compensation

Table 3 summarises the compensation offered to participants in the 14 trials that reported on compensation. The type of compensation differed among trials, but half reported using an e-gift card or e-voucher (n = 7). Five trials did not report the form of compensation, only the amount, and two offered a chance to win an e-voucher or other prize. Compensation ranged from a US $3.50 one-time payment to US $120 to complete all study requirements. None of the trials that required caregiver consent or included data submitted by both caregivers and children specified who received the compensation.

## Loss to follow-up

The trials included in this scoping review reported attrition differently and often with differing terminology. The attrition rates can be found in the complete RoB tables in S3 Appendix.

Single-session interventions accounted for seven of the 21 trials (most likely eight, however, Hillhouse et al. [36] did not report the duration of their intervention), and six of those (including Hillhouse et al.) collected follow-up data. The remaining 13 trials varied in duration, and nine collected follow-up data. Arnaud et al. [29] had the highest loss to follow-up rate (intervention, 84.9% & comparison, 86%) of all included trials. This trial had the third largest number of participants (n = 1,449), was a single-session intervention, and had a 3-month follow-up assessment. The authors hypothesised that this high dropout rate might be due to either invalid email addresses provided by participants or the use of only one reminder email sent to each participant [29]. Bragg et al. [30] was a single-session intervention with no follow-up data requirements conducted with 832 participants, offered no compensation, but reported the lowest loss to follow-up (reported as 6.5% across combined intervention and comparator arms). Twelve trials sent reminders to participants [29,31,33–34,38,39,41–44,46,47] for enrolment, retention, or follow-up data collection.

## Children's input in the design or conduct of trials

Thirty-eight percent of the 21 trials (n = 8) collected input from children to inform their trial design or conduct [33,35,36,38,42–45]. Half of these trials reported conducting pilot tests of

**Table 2. Online platforms and specific social media platforms used for recruitment to trials.**

| Study | Online platforms used for recruitment | Social media platform(s) used for recruitment |
|---|---|---|
| Amsalem and Martin, 2022 [28] | External Recruitment Panel | No |
| Arnaud et al., 2016 [29] | Multiple | Yes, platform not reported |
| Bragg et al., 2021 [30] | External Recruitment Panel | No |
| Craig et al., 2016 [31] | Multiple | Yes, platform not reported |
| Dobias et al., 2021 [32] | Social media | IG |
| Egan et al., 2021 [33] | Multiple | FB, IG |
| Ghaderi et al., 2020 [34] | Multiple | FB, IG |
| Greene et al., 2020 [35] | Multiple | FB |
| Hillhouse et al., 2017 [36] | External Recruitment Panel | No |
| Kelleher, Moreno, & Wilt, 2018 [37] | Social media | TR |
| Lester et al., 2019 [38] | Email | No |
| Manicavasagar et al., 2014 [39] | Multiple | No |
| Mogil et al., 2021 [40] | Social media | Yes, platform not reported |
| Moreno et al., 2021 [41] | External Recruitment Panel | No |
| Nelson et al., 2022 [42] | Social media | FB, IG |
| O'Connor et al., 2020 [43] | Multiple | FB, IG, TW |
| O'Dea et al., 2020 [44] | Social media | FB |
| Parker, Scull, & Morrison, 2022 [45] | Social media | Yes, platform not reported |
| Radomski et al., 2020 [46] | Social media | FB, IG, TR, TW |
| Schleider et al., 2022 [47] | Social media | IG |
| Schwinn et al., 2015 [48] | Social media | FB |

FB = Facebook, IG = Instagram, TR = Tumblr, TW = Twitter.

their interventions for participants' feedback. Only one trial [35] reported how they recruited their PPI group and provided their ages. Two trials [43,45] collected input from clinicians and participants, and Parker, Scull & Morrison [45] further collected feedback from parents. Egan et al. [33] interviewed 20 youths from their target population to inform their trial design. Nelson et al. [42] assembled a youth advisory group and conducted a cross-sectional survey with their target demographic to inform the design and content of their intervention website. None of the trials discussed PPI compensation.

## Risk of bias (RoB) assessment

The complete RoB tables for each trial are available in S3 Appendix. Fig 2 displays the RoB summary data. The three domains where most assessments were 'low' risk were "random sequence generation" (n = 12), "selective reporting" (n = 14), and "other" bias (n = 17). 'Unclear' risk assessments were most common in "allocation concealment" (n = 11). The three domains with the most 'high' risk assessments were "blinding of participants and personnel" (n = 13), "blinding of outcome assessment" (n = 11), and "incomplete outcome data" (n = 11). The trial with the least risk of bias assessments was Schleider et al. [47] with six 'low' risk assessments. Four trials had five 'low' risk assessments across domains (Arnaud et al. [29], Ghaderi et al. [34], Nelson et al. [42], and O'Dea et al. [44]). The trial with the most (n = 5) uncertainty in risk of bias across domains was Schwinn et al. [48]. The trials with the highest risk of bias across domains (n = 5) were Manicavasagar et al. [39] and O'Connor et al. [43].

**Table 3. Compensation type and amounts reported.**

| Study | Form of compensation offered | Amount/ gift/ benefit offered |
|---|---|---|
| Amsalem and Martin, 2022 [28] | NR | US $3.50 |
| Arnaud et al., 2016 [29] | Prize draw competition | Tablet computer |
| Dobias et al., 2021 [32] | Cash prize draw competition | 1-in-10 chance of winning US $25 at baseline, US $5 guaranteed at follow-up |
| Egan et al., 2021 [33] | NR | US $10 at baseline, US $25 at post-test, US $50 at follow-up |
| Greene et al., 2020 [35] | NR | US $10 at baseline, US $10 at post-test, US $10 at follow-up |
| Hillhouse et al., 2017 [36] | NR | up to US $120 for completing all study requirements |
| Kelleher, Moreno, & Wilt, 2018 [37] | E-gift card/voucher | US $10 at survey completion |
| Manicavasagar et al., 2014 [39] | E-gift card/voucher | AU $20 at study completion |
| Nelson et al., 2022 [42] | E-gift card/voucher | US $15 at baseline, US $25 at post-test, US $35 at follow-up, US $20 bonus for all three assessments |
| O'Dea et al., 2020 [44] | E-gift card/voucher | AU $15 at baseline, AU $15 at post-test, AU $15 at follow-up |
| Parker, Scull, & Morrison, 2022 [45] | NR | US $20 at baseline, US $30 at follow-up |
| Radomski et al., 2020 [46] | E-gift card/voucher | CA $25 |
| Schleider et al., 2022 [47] | E-gift card/voucher | Up to US $20 |
| Schwinn et al., 2015 [48] | E-gift card/voucher | US $25 at baseline, US $30 at post-test, US $45 at follow-up |

NR = Not reported.

## Discussion

We identified 21 trials with children conducted entirely online that met our inclusion criteria from 6,957 screened, highlighting the scarcity of published research in this area. We found that the reporting of recruitment, retention and public involvement methods in the included studies were not always sufficient to assess or replicate. Consent acquisition, data collection, and compensation methods were reported more thoroughly.

The use of social media as a recruitment tool was most prevalent among the studies, with specific studies noting that this approach significantly sped up their recruitment process [33,34,43,44,47]. However, apart from these five trials, there was no distinct information regarding the effectiveness of the recruitment methods used in the other studies. To facilitate better planning and implementation in future research, it would be advantageous if researchers could include details on the efficiency of their recruitment strategies in their reports, whenever feasible. Ten trials created dedicated study websites, but only three [29,34,35] reported using them in their recruitment efforts. Newly created websites only receive traffic from word-of-mouth, seed recruitment (where one participant is responsible for recruiting other participants), advertising via other websites or search engines, and offline media. Relying on dedicated study websites or emailing listservs exclusively generated the least amount of enrolment. Four trials used commercial recruitment platforms, which advertise that they can recruit rapidly and cost-effectively, but there are concerns they lack transparency on cost, bias, and potential over-use of a narrow participant pool [9,15].

A recent scoping review of 33 studies (1/33 was with children) examining recruitment to clinical trials via social media, traditional methods, and other online media reported that seven out of 20 trials had the highest recruitment rates with social media compared to other methods [56]. Since most social media platforms have a lower age limit of 13 [57], and the digital age of consent varies across countries [58], researchers need to consider both the legal and ethical implications of targeting social media recruitment toward children. Five trials in our review included children under 13 and reported using social media for recruitment [31,35,40,44,45].

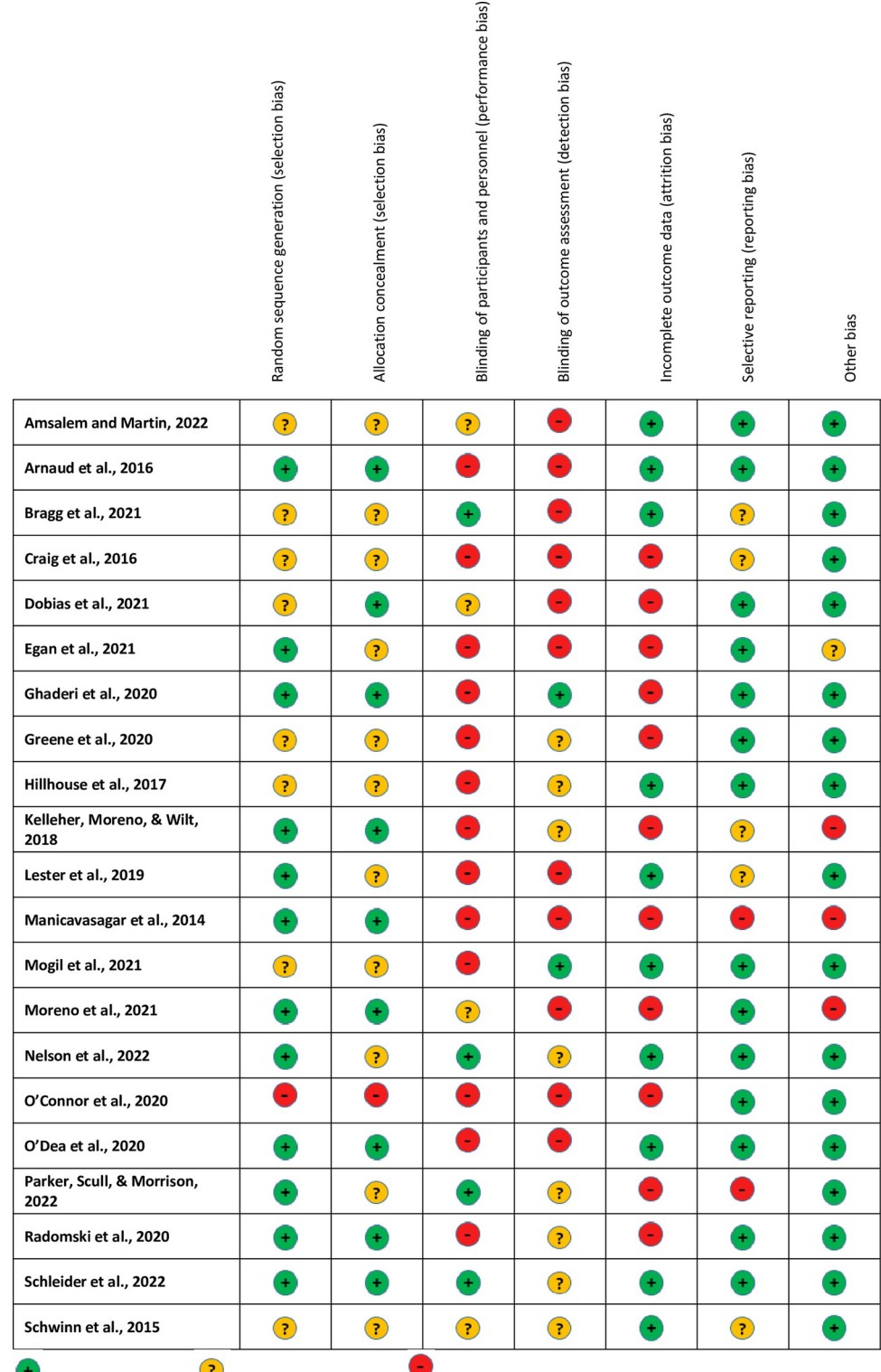

Fig 2. Risk of bias (RoB) assessment summary for all included trials.

Three of these trials [31,35,40] reported targeting parents on social media, and two [44,45] did not. Considering that all studies in our review received ethical approval, it is possible that social media advertising was targeted toward caregivers, but we are unable to make that assessment. Social media is an effective recruitment tool to reach adolescents who have higher social media usage than older adults [59], but not always appropriate for all paediatric populations. We identified a gap in the knowledge in the reporting of recruitment methods by many of the studies included in this review. There is a need for researchers to outline their recruitment plans and recruitment processes in more detail to contribute to the knowledge base.

Very little was reported on retention methods, and no analyses were done on the participant retention rates and engagement efforts. Online trials are often less burdensome for participants as they do not impose in-person interactions that may require travel, increased time commitments, and scheduling problems. This could result in higher participant retention; however, based on this review's findings, we cannot tease out which methods used to engage and retain child participants might have most potential. Delineating recruitment and retention in trials can be challenging. When a website or app is part of a trial, their ease of use and visual impression affects how participants view the trial. Because of this, the technology and methods used for recruitment and delivering the trial intervention are linked intricately with a trial's retention rates. There were trials in our study that had high-quality data and high retention of participants showing that choosing suitable strategies to undertake online trials with children is possible and appropriate. Nonetheless, there is a noticeable gap in the literature regarding the specific methods that facilitated high retention rates in these studies. Identifying and understanding these facilitators is crucial to enhancing future research and improving participant retention.

Our review found a lack of PPI input, and that input was not always sufficiently reported. Of the eight trials that sought feedback or input from children in the design of their trials, only one reported on how they recruited for their PPI group. Online advisory groups may be difficult to navigate logistically and therefore present a challenge to trial design; however, the involvement of PPI is key to increasing inclusivity and relevance of trial outcomes [60]. Further research is needed on how online children's PPI groups are best compensated so that it can be integrated into trial design.

The trials in our review did not report on cost analyses or comparisons, so we cannot make any assessments regarding the cost of different recruitment methods in online trials with children and have therefore identified this as another gap in the knowledge. Cost comparisons would be useful for trialists, although it may be difficult to determine if the cost of different online recruitment and retention methods affect those outcomes. However, beyond relying on the public, professional and advocacy groups to 'share' the trial details, all social media advertising incurs a cost, as do advertisements on search engines, websites, and the use of external recruitment panels. In two recent reviews that examined cost-effectiveness of online recruitment methods [56,61], social media had a lower cost per enrolee than traditional enrolment methods.

Consent invitation processes varied, with e-consent links emailed or sent direct messages via social media to interested or potentially interested participants as the most common tool. Automated re-direction links to external websites were the second most common method. Ten trials [30–32,35,38,40,42,43,46,48] in our review created consent validation checks demonstrating creative and varied methods to increase their participants' comprehension of the trial. Nelson et al. [42] confirmed capacity to consent using questions that asked participants to identify what the study was expecting of them, if they understood how their group allocation was decided, how to handle any distress they might experience, and to identify what risks might be present during the study. O'Connor et al. [43] added True/False questions about the

study before consent/assent was given and incorrect answers activated a pop-up box that explained the correct answer.

A 2020 scoping review of 69 papers describing electronic informed consent in medical centres (three paediatric trials, one trial with both children and adults, 65 with adults) found no universal approach to gaining consent online, and the legal requirements for consent by country or region varied [62]. Nine of the 69 trials found that participants understood online consent more easily than traditional, paper consent forms and that online or e-consent forms may present a more streamlined approach in paediatric trials because in-person parental consent appointments often challenge parents and researchers [62].

Data collection methods were reported thoroughly in most of the trials included in our review. No challenges were reported in achieving adequate data protection measures, and external host websites appeared to fit this need for several studies. Where studies addressed fraudulent or duplicate entries (when a participant submits data or enters the trial more than once for incentive purposes or purely by mistake) this was addressed comprehensively and creatively, demonstrating several ways to decrease the likelihood of their occurrence [28,30,32,37]. Amsalem et al. [28] combined CAPTCHA questions, timers on questions and videos, and scanned for repeat or suspicious IP addresses. Bragg et al. [30] checked participants' attentiveness by asking them to type a word into a box and excluded incorrect answers. Dobias et al. [32] required participants to answer data-quality questions that checked for attentiveness, English fluency, and at least three-word answers. Kelleher et al. checked for repeat IP addresses to identify multiple entries to their study [37]. These methods likely facilitated a decrease in the incidence of fraudulent and duplicate entries.

No barriers were reported regarding the online compensation of participants. E-gift cards or vouchers were the most common and acceptable compensation in the trials included here.

Based on the findings of this review, we have collated the following considerations when designing and conducting online trials with children:

- social media and 'other' online advertising expedite recruitment efforts in these populations, especially if parents or caregivers are the target recruitment pool

- use social media for paediatric populations above the age of 13

- combine methods of online recruitment to widen the participant pool, i.e. search engine advertising, social media advertising, school-, community-, healthcare facility-, and child-centric- website advertising

- consider conducting a SWAT (Study Within A Trial [63]) to help determine which recruitment methods best fit the needs of the trial

- consider transparency of sampling, randomisation and recruitment methods if employing commercial recruitment platforms

- involve PPI input as early as possible

- PPI group recruitment and involvement for online trials should occur online

- PPI group meetings should accommodate both children and parents'/caregivers' schedules

- implement appropriate compensation for PPI partners, consider hourly rates for both parents/caregivers keeping local tax and labour laws in mind

- embed consent validation methods to ensure informed consent/assent (e.g., questions to participants to check for understanding, using timers on screens to encourage all information has been read or heard, double-checking IP addresses to decrease duplicate entries)

- check country and regional laws for online consent guidance

- fit-for-purpose external host websites store data securely and may assist in efficient data collection.

### Strengths and limitations

We used a robust search strategy across a broad scope of literature sources. We did not limit our searches by date and applied a quality assessment of included studies completed independently by two reviewers in duplicate. We conducted our screenings and data charting in a reproducible and transferable manner. We assessed the included studies using a practical approach to understand and inform key stakeholders of the gaps in the knowledge, as well as which methods used in designing and conducting online trials with children were effective.

We limited our search to the English language publications because translation services would require additional resources. After the pilot screenings of the pre-determined number of abstracts and full texts, one author completed the remainder of those processes. While this provides consistency, independent duplicate screening may be preferable.

Two authors piloted the data charting of five included studies to determine the levels of agreement, and one author charted the remaining studies' data. Because the trials were so heterogeneous, some comparisons between trials were not possible. This was an anticipated and acceptable limitation since the methodologies were of greater interest than the trial outcomes. This scoping review's sole focus is online, randomised trials with children to identify and describe the methods and tools used to design and conduct these trials. We acknowledge there remains a large digital divide through either economic disadvantage, infrastructure availability or personal choice. Nearly half of the world's population remains offline [64], creating an entirely different ethical dilemma when conducting these trials.

### Conclusions

The methods, facilitators, and gaps in the knowledge of how online, randomised trials with children are conducted were identified and described; most trials used social media for multiple phases of their trials, although this may not always be the most suitable platform for these methods. More in-depth consideration of target age groups, population, and PPI input should inform the design and conduct of these trials, and more thorough reporting of the methods is needed in future studies.

### Supporting information

**S1 Checklist. Preferred Reporting Items for Systematic Reviews and Meta-Analyses Extension for Scoping Reviews (PRISMA-ScR) checklist.**
(DOCX)

**S1 Table. Racial and ethnic minority distribution of participants by author and country in "Online randomised trials with children: A scoping review".** Legend: NR = Not reported; NA = Not applicable. * = Data taken from parents, maternal data is listed first, paternal data is listed second. $ = Aboriginal or Torres Straight Islander. Where percentages do not equal 100, it was generally a case that Hispanic or Latino were not mutually exclusive from White/Caucasian as they can be considered an ethnicity and/or heritage, and the individual's race may be listed differently. An individual may be Hispanic/ Latino and Black or Hispanic/Latino and

White, etc., they also may list their ethnicity only as Hispanic/Latino.
(DOCX)

**S2 Table. Recruitment methods of participants reported for "Online randomised trials with children: A scoping review".**
(DOCX)

**S1 Appendix. Database search strategies developed for MEDLINE (Ovid), CENTRAL, CINAHL (EBSCO), & Embase (Elsevier) for "Online randomised trials with children: A scoping review".**
(DOCX)

**S2 Appendix. Grey literature summary of search terms and strings used in grey literature searches conducted for "Online randomised trials with children: A scoping review".**
(DOCX)

**S3 Appendix. Completed data charting and risk of bias assessments for all included studies.**
(ZIP)

**S1 Form. Data charting tool for "Online randomised trials with children: A scoping review".**
(DOCX)

## Acknowledgments

This work contributes to one of the authors' (SL) doctoral projects. The findings and conclusions in the document are those of the authors and not necessarily those of the authors employing organisations.

## Author Contributions

**Conceptualization:** Simone Lepage, Declan Devane.

**Data curation:** Simone Lepage, Declan Devane.

**Formal analysis:** Simone Lepage, Declan Devane.

**Funding acquisition:** Declan Devane.

**Investigation:** Simone Lepage, Laura Flight, Declan Devane.

**Methodology:** Simone Lepage, Aislinn Conway, Noah Goodson, Paul Wicks, Laura Flight, Declan Devane.

**Project administration:** Simone Lepage, Declan Devane.

**Supervision:** Laura Flight, Declan Devane.

**Validation:** Simone Lepage, Aislinn Conway, Noah Goodson, Paul Wicks, Laura Flight, Declan Devane.

**Visualization:** Simone Lepage, Declan Devane.

**Writing – original draft:** Simone Lepage, Declan Devane.

**Writing – review & editing:** Simone Lepage, Aislinn Conway, Noah Goodson, Paul Wicks, Laura Flight, Declan Devane.

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
