## [Decision Letter · Decision Letter 0]

26 Apr 2023

PONE-D-23-00748Online randomised trials with children: A scoping reviewPLOS ONE

Dear Dr. Lepage,

Thank you for submitting your manuscript to PLOS ONE. After careful consideration, we feel that it has merit but does not fully meet PLOS ONE’s publication criteria as it currently stands. Therefore, we invite you to submit a revised version of the manuscript that addresses the points raised during the review process.

ACADEMIC EDITOR: Thanks for the opportunity to review this paper. This scoping review aimed to identify methods and tools have been used to create and conduct entirely online decentralized trials with children and determine knowledge gaps.

This review addresses important topic in the field and may interest to researchers who work in paediatric fields. Overall it is well written. Several points were provided for considerations when designing and conducting online trials with children. I agreed with other reviewers that more elaboration on facilitators and gaps in knowledge as well as implications of this review should be added.

We look forward to receiving your revised manuscript.

Kind regards,

Cho Lee Wong, PhD

Academic Editor

PLOS ONE

Journal Requirements:

"This work contributes to one of the authors’ (SL) doctoral projects. SL’s PhD is funded by the College of Medicine, Nursing and Health Sciences at the University of Galway and by the Health Research Board (HRB, Ireland) through a grant to the Health Research Board-Trials Methodology Research Network (HRB-TMRN). "

"This research was supported by the Health Research Board [TMRN-2021-001] and by the College of Medicine, Nursing and Health Sciences at the University of  Galway. Funding was received by DD from the Health Research Board of Ireland: https://www.hrb.ie/

The funder had no role in study design, data collection and analysis, decision to publish, or preparation of the manuscript."

"PW is an associate editor at the Journal of Medical Internet Research and is on the editorial advisory boards of The BMJ, BMC Medicine, The Patient, and Digital Biomarkers. PW is employed by Wicks Digital Health Ltd, which has received funding from Ada Health, AstraZeneca, Biogen, Bold Health, Camoni, Compass Pathways, Coronna, EIT, Endava, Happify, HealthUnlocked, Inbeeo, Kheiron Medical, Lindus Health, MedRhythms, PatientsLikeMe, Sano Genetics, Self Care Catalysts, The Learning Corp, The Wellcome Trust, THREAD Research, VeraSci, and Woebot.

NG is an employee of THREAD, a decentralized platform provider.

The remaining authors, AC, DD, LF, and SL have no conflicts of interest to report.

The findings and conclusions in the document are those of the authors and not necessarily those of NICE."

Additional Editor Comments:

Thanks for the opportunity to review this paper. This scoping review aimed to identify methods and tools have been used to create and conduct entirely online decentralized trials with children and determine knowledge gaps.

This review addresses important topic in the field and may interest to researchers who work in paediatric fields. Overall it is well written. Several points were provided for considerations when designing and conducting online trials with children. I agreed with other reviewers that more elaboration on facilitators and gaps in knowledge as well as implications of this review should be added.

Reviewers' comments:

Reviewer's Responses to Questions

**Comments to the Author**

1. Is the manuscript technically sound, and do the data support the conclusions?

Reviewer #1: Yes

Reviewer #2: Partly

2. Has the statistical analysis been performed appropriately and rigorously? 

Reviewer #1: Yes

Reviewer #2: N/A

3. Have the authors made all data underlying the findings in their manuscript fully available?

Reviewer #1: Yes

Reviewer #2: No

4. Is the manuscript presented in an intelligible fashion and written in standard English?

Reviewer #1: Yes

Reviewer #2: Yes

5. Review Comments to the Author

Reviewer #1: As scoping reviews do not aim to produce a critically appraised and synthesised result/answer to a particular question, and rather aim to provide an overview or map of the evidence, an assessment of methodological limitations or risk of bias of the evidence included within a scoping review is generally not performed (unless there is a specific requirement due to the nature of the scoping review aim). In this paper, the aim was to evaluate fully online RCTs performed on children, so that the authors could identify the methods and tools used, and to determine the gaps in knowledge, in order to help them perform their own online trial in children.

The methods used were thorough, necessary, and well documented. They looked at a large number of trials, and followed the protocol meticulously. They were left with 21 studies that met with all the criteria. Exclusion of other studies was well justified. What bothers me about online trials is bias in recruitment, accuracy and validity of the information gathered, and spurious results. Did the authors find ways in which the authors of the different studies accounted for this? Also, were there ways in which the authors of the studies gauged the comprehension of the participants of the tasks set out for them? For informed consent to be valid we need the recruits to understand the implications of their participation. In addition, it would be preferable to include all studies in languages other than English, so as to be more comprehensively cover the subject of study. I understand that lack of funds precluded the authors to include other languages studies, but that does not totally justify excluding them. I agree with the authors’ conclusion that more than one person should have screened all the studies, which would have provided more valid representation of the studies.

I agree with the conclusions of the review.

Reviewer #2: This study presented a research question in summarising the evidence in the online RCTs in children. However, many issues should be addressed first before the manuscript can be considered for publication, especially the research problem clarification. My suggestions are as below:

Introduction:

1. Please reconsider the research question and make it more clearly. What specific online RCT did the authors want to evaluate. Which type of the children did the authors want to focus on. What specific problems of children did the authors want to address. After that, please also rewrite the title and introduction.

2. As only RCTs or quasi experimental studies were included, why a scoping review was still conducted, not a systematic review or meta-analysis?

Methods:

1. After reclarifying the research problem, the authors should redefine the inclusion criteria of the studies

2. How about the quality appraisal process? Please add more details of ROB

Results:

1. The authors seemed to write this review as a wrong direction. Actually, for the recruitment methods, consent, data collection, compensation, loss to follow-up, and public and patient involvement (PPI) in online trials within paediatric populations, these could be summarised as the quality appraisal results, but should not be the main focus of a systematic review. The main results of the review involving RCTs should summarised the components of the target intervention, such as intervention typer, mode, frequency, and the efficacy of the interventions.

2. What facilitators and gaps did the authors find out?

Discussion:

1. The authors should reconsider the significance of the findings for international scholars.

6. PLOS authors have the option to publish the peer review history of their article (what does this mean?). If published, this will include your full peer review and any attached files.

Reviewer #1: No

Reviewer #2: No

---

## [Author Response · Author response to Decision Letter 0]

10 May 2023

Dear Dr Wong,

We would like to extend our thanks and appreciation to all the reviewers for their commentary and feedback. This letter provides our response to the reviewers’ comments to our manuscript, “Online randomised trials with children: A scoping review”. 

Journal Requirements:

1. The manuscript file names comply with PLOS ONE style requirements.

2. Acknowledgment Section & Funding Statement

The Acknowledgements Section has been amended and no longer contains any funding information.

The Funding Statement should read: 

“This research was supported by the Health Research Board [TMRN-2021-001] and by the College of Medicine, Nursing and Health Sciences at the University of Galway. Funding was received by DD from the Health Research Board of Ireland: https://www.hrb.ie/. The funder had no role in study design, data collection and analysis, decision to publish, or preparation of the manuscript.”

We have amended our cover letter appropriately.

3. Competing Interests Section

Our Competing Interests Section remains the same and reads as follows: 

“PW is an associate editor at the Journal of Medical Internet Research and is on the editorial advisory boards of the BMJ, BMC Medicine, The Patient, and Digital Biomarkers. PW is employed by Wicks Digital Health Ltd, which has received funding from Ada Health, AstraZeneca, Biogen, Bold Health, Camoni, Compass Pathways, Coronna, EIT, Endava, Happify, HealthUnlocked, Inbeeo, Kheiron Medical, Lindus Health, MedRhythms, PatientsLikeMe, Sano Genetics, Self Care Catalysts, The Learning Corp, The Wellcome Trust, THREAD Research, VeraSci, and Woebot.

NG is an employee of THREAD, a decentralised platform provider.

The remaining authors, AC, DD, LF, and SL have no conflicts of interest to report.

The findings and conclusions in the document are those of the authors and not necessarily those of NICE.” 

However, we would like the following statement added: 

“This does not alter our adherence to PLOS ONE policies on sharing data and materials.” 

Points raised by the academic reviewer:

1. “More elaboration on the facilitators and gaps in knowledge and their implications should be added.”

Our response: We have added more detail in the Discussion Section of the manuscript addressing the facilitators and gaps in the field and their implications.

Points raised by Reviewer #1:

1. “As scoping reviews do not aim to produce a critically appraised and synthesised result/answer to a particular question, and rather aim to provide an overview or map of the evidence, an assessment of methodological limitations or risk of bias of the evidence included within a scoping review is generally not performed (unless there is a specific requirement due to the nature of the scoping review aim).” 

Our response: We acknowledge and appreciate that a Risk of Bias assessment is not customary in a scoping review, nor is it necessary; however, we believe its inclusion in our review enhances the overall value and utility of our findings. By evaluating the quality of the included studies, readers gain a clearer understanding of the evidence's reliability, which in turn, supports the development of more robust research methodologies in future studies.

. 

2. “What bothers me about online trials is bias in recruitment, accuracy and validity of the information gathered, and spurious results. Did the authors find ways in which the authors of the different studies accounted for this? Also, were there ways in which the authors of the studies gauged the comprehension of the participants of the tasks set out for them? For informed consent to be valid we need the recruits to understand the implications of their participation.”

Our response: The concerns surrounding decentralised, online trials raised by Reviewer #1 are echoed in the literature and although we did address this in the Introduction and Discussion Sections, we have amended the manuscript to elaborate on these concerns. We hope our recommendations will guide researchers so that these challenges can be met in a more rigorous and transparent manner. 

3. “In addition, it would be preferable to include all studies in languages other than English, so as to be more comprehensively cover the subject of study. I understand that lack of funds precluded the authors to include other languages studies, but that does not totally justify excluding them. “ 

Our response: The concern of including only English-language studies is a relevant one. This is a challenge that many researchers face when working within their budgets. We acknowledge that this is a limitation of our review and introduces bias to our findings. Unfortunately, we are unable to change our resource allocation at this stage of the process. 

Points raised by Reviewer #2:

Comments to the Author Responses

Question 3 posed to the reviewers: Have the authors made all data underlying the findings in their manuscript fully available?

Reviewer #2 answered “No”. 

Our response: All underlying data are available either in the associated supporting documentation as well as on the Open Science Framework website: http://www.doi.org/10.17605/OSF.IO/WHJXY

Introduction

1. “Please reconsider the research question and make it more clearly. What specific online RCT did the authors want to evaluate. Which type of the children did the authors want to focus on. What specific problems of children did the authors want to address. After that, please also rewrite the title and introduction. 

2. “As only RCTs or quasi experimental studies were included, why a scoping review was still conducted, not a systematic review or meta-analysis?”

Our response: We will address points 1 and 2 together. As discussed in our Introduction Section and echoed by both the Academic Reviewer and Reviewer #1, the purpose of conducting a scoping review was to determine what the gaps in the knowledge of this field are and identify what the methods and tools were used in entirely online, decentralised trials within paediatric populations. The aim of a systematic review is to guide a decision-making process or answer a focused research question, which was not our aim with this manuscript. Our reasoning for conducting a scoping review was to inform the planning and conduction of our own online, randomised controlled trial with children. Because of this, a scoping review was more appropriate. Our primary focus was not on the outcomes of the interventions; instead, we were more interested in examining the trial methodologies. This approach allowed us to gain valuable insights into the study designs and their implications on the research process. This is also the reason a meta-analysis was not conducted. We were less interested in the results of interventions, rather the methodologies of the trials were of interest. 

Methods

1. “After reclarifying the research problem, the authors should redefine the inclusion criteria of the studies.”

Our response: As addressed above. The inclusion and exclusion criteria were established to identify studies that were either a quasi-randomised or a randomised trial, conducted in a paediatric population, primarily employing online methods for the majority of their procedures, and published in English.

2. “How about the quality appraisal process? Please add more details of ROB.”

Our response: The full RoB assessments with justifications for decisions made are available in the supporting document S7 Appendix. 

Results

1. “The authors seemed to write this review as a wrong direction. Actually, for the recruitment methods, consent, data collection, compensation, loss to follow-up, and public and patient involvement (PPI) in online trials within paediatric populations, these could be summarised as the quality appraisal results, but should not be the main focus of a systematic review. The main results of the review involving RCTs should summarise the components of the target intervention, such as intervention type, mode, frequency, and the efficacy of the interventions. “

Our response: As addressed above under Introduction points 1 & 2 responses. 

2. “What facilitators and gaps did the authors find out?”

Our response: We have amended the Discussion section to clarify facilitators and gaps in the knowledge. 

Discussion

1. “The authors should reconsider the significance of the findings for international scholars.”

Our response: Our aim in the recommendations made in the Discussion Section is that they are applicable to researchers in most countries and geographical areas. 

Again, we are thankful for the feedback provided by the reviewers of this manuscript.

---

## [Editor Report · Decision Letter 1]

11 May 2023

Online randomised trials with children: A scoping review

PONE-D-23-00748R1

Dear Simone Lepage,

We’re pleased to inform you that your manuscript has been judged scientifically suitable for publication and will be formally accepted for publication once it meets all outstanding technical requirements.

Kind regards,

Cho Lee Wong, PhD

Academic Editor

PLOS ONE
---

## [Editor Report · Acceptance letter]

15 May 2023

PONE-D-23-00748R1 

Online randomised trials with children: A scoping review 

Dear Dr. Lepage:

I'm pleased to inform you that your manuscript has been deemed suitable for publication in PLOS ONE. Congratulations! Your manuscript is now with our production department. 

Kind regards, 

on behalf of

Dr. Cho Lee Wong 

Academic Editor

PLOS ONE